# The Yeast Ribosomal Protein Rpl1b Is Not Required for Respiration

**DOI:** 10.3390/ijms252111553

**Published:** 2024-10-28

**Authors:** Bruce Futcher

**Affiliations:** Department of Microbiology and Immunology, Stony Brook University, Stony Brook, NY 11794, USA; bfutcher@gmail.com

**Keywords:** petite, respiration, ribosomal proteins, ribosomal paralogs, Rpl1b

## Abstract

Previously, Segev and Gerst found that mutants in any of the four ribosomal protein genes *rpl1b*, *rpl2b*, *rps11a*, or *rps26b* had a petite phenotype—i.e., the mutants were deficient in respiration. Strikingly, mutants of their paralogs *rpl1a*, *rpl2a*, *rps11b*, and *rps26a* were grande—i.e., competent for respiration. It is remarkable that these paralogs should have opposite phenotypes, because three of the paralog pairs (Rpl1a/Rpl1b, Rpl2a/Rpl2b, Rps11a/Rps11b) are 100% identical to each other in terms of their amino acid sequences, while Rps26a and Rps26b differ in 2 amino acids out of 119. However, while attempting to use this paralog-specific petite phenotype in an unrelated experiment, I found that the *rpl1b*, *rpl2b*, *rps11a*, and *rps26b* deletion mutants are competent for respiration, contrary to the findings of Segev and Gerst.

## 1. Introduction

In yeast, many ribosomal proteins are encoded by a pair of paralogous genes, e.g., *RPL1A* and *RPL1B*. Often, one of the paralogs can be deleted to yield a viable but often slowly growing cell, while deleting both paralogs is lethal. The two paralogs are not necessarily expressed equally, so one paralog deletion may have a stronger phenotype than the other. In some cases, the two paralogs encode identical proteins (e.g., Rpl1a and Rpl1b, 217/217 identical amino acids), while in others, there are small differences between the proteins (e.g., Rps26a and Rps26b, 117/119 identical amino acids).

Because some ribosomal proteins differ in sequence from their paralogs, it follows that some ribosomes are different from others, depending on which paralog they contain. There has been interest in whether these different ribosomes have different, perhaps specialized, functions [1,2,3,4,5].

A striking, surprising, and highly cited argument for ribosome specialization has come from Segev and Gerst [6], who found that *rpl1b* mutants, and also *rpl2b*, *rps11a*, and *rps26b* mutants, were respiratory-deficient (i.e., petite), while the paralogous mutants (*rpl1a*, *rpl2a*, *rps11b*, *rps26a*) were not. Segev and Gerst argue this difference cannot be explained by differences in the expressions of *RPL1a* and *RPL1b*. What is particularly striking about this finding is that Rpl1a and Rpl1b have identical amino acid sequences, so if there is a qualitative difference in function, it must somehow lie in non-coding regions (which differ in DNA sequence), or, conceivably, in the two nucleotides in the open reading frame that are different but do not alter the amino acid sequence. It is unclear what mechanisms could make Rpl1a-containing ribosomes different from Rpl1b-containing ribosomes when the two encoded proteins are identical, but Segev and Gerst discuss possible models.

However, when attempting to use this paralog-specific petite phenotype as an assay in an unrelated experiment, I found that none of the four mutants (*rpl1b*, *rpl2b*, *rps11a*, or *rps26b*) were petite. Instead, all four were respiratory-proficient, i.e., grande.

## 2. Results

The BY4742 *MATalpha rpl1b* strain was taken from the yeast deletion collection, and crossed to the isogenic strain BY4741, yielding diploid BF534 (Methods and Materials). Surprisingly, upon tetrad dissection, G418 resistance segregated 3:1 or 4:0, suggesting that the strain from the deletion set contained two (unlinked) copies of the KanR gene instead of one. Tetrad 12, which segregated four G418R:0 sensitive, was chosen for further analysis. In tetrad 12, spore clones A and C were fast-growing, while spore clones B and D were slow growing. PCR showed that all four spore clones contained *KANR::rpl1b* (i.e., *KANR* replacing the open reading frame of *RPL1B*), while spore clones A and C (but not B and D) also contained a wild-type copy of *RPL1B* (Appendix A). This suggests that the *MATalpha rpl1b* strain in the deletion set is a disome for chromosome VII, with both copies of the chromosome containing *KANR::rpl1b*. (Segev and Gerst made their own deletions of *rpl1b*, and did not use this deletion set strain [6]) (see Appendix A for a representation of disome segregation).

Although the A and C spore clones are presumably disomic chromosome VII, *RPL1B/rpl1b*, nevertheless, the slow-growing B and D spore clones seem to be the desired *rpl1b* mutants. (Below, to confirm this, spore 12D was crossed to wild-type strain GZ239, and G418 resistance and slow growth and the *KANR::rpl1b* deletion co-segregate with each other 2:2, as expected.)

To create “gold-standard” petite controls, the 12A, B, C, and D spore clones were grown in the presence of ethidium bromide (Materials and Methods), which reliably causes a loss of mitochondrial DNA and converts grandes into petites [7,8]. Thus, each of the four spore clones (whether these were originally petite or not) had a known petite version generated by growth in ethidium bromide. As shown below, these control petites generated using ethidium bromide had a complete defect in respiration.

### 2.1. Phenotypes of the rpl1b Mutants BF534-12B and 12D

BF534-12B and 12D were both slow-growing on YEPD, as expected for an *rpl1b* deletion, or for a petite. They both also formed the bone-white colonies typical of a petite, instead of the creamier colonies of a grande.

### 2.2. The rpl1b Deletion Mutants Grow on YGlyE Plates

A common indicator medium for petites is YPdG (1% yeast extract, 2% peptone, 0.1% glucose, 3% glycerol), on which petites form very tiny colonies, and grandes form large colonies [9]. I used a version of this medium containing ethanol as well as glycerol (YGlyE, Materials and Methods) for an initial test of whether strains were petite or grande. As shown in Figure 1, all four spore clones of tetrad 12, whether they had the WT *RPL1B* gene or not, grew quite well on YGlyE, while the ethidium bromide that generated petite controls grew slowly, as expected. (Note the figure shows YEPD plates after 2 days of growth, and YGlyE plates after 4 days.) The 12B and 12D spore clones grew somewhat more slowly than the 12A clone, but this was also true on YPD (i.e., with glucose as a carbon source), and it is expected for *rpl1b* null mutants. Thus, as assayed by YGlyE plates, the *rpl1b* null mutants are grande, not petite.

### 2.3. The rpl1b Deletion Mutants Grow in Synthetic Medium with Ethanol as the Carbon Source

12A, B, C, and D, and their petite controls generated using ethidium bromide, were inoculated into 5 mL of Yeast Nitrogen Base synthetic medium, with only the required amino acids, 2% ethanol, and 0.02% glucose. The 0.02% glucose is present because even wild-type strains can fail to proliferate when switched abruptly from glucose medium to YNB–ethanol medium [10]. A small amount of glucose allows cells the resources to switch from fermentative to respirative growth. As controls, cells were also inoculated into the same medium with 2% glucose as a carbon source, or into the same medium with only 0.02% glucose.

As shown in Figure 2, 12A, B, C, and D all grew to high density (~1 × 10^8^/mL) in the 2% glucose and the 2% ethanol + 0.02% glucose medium, but did not grow significantly in the 0.02% glucose medium. In contrast, the four petite versions grew to high density only in the 2% glucose medium, but not in the 2% ethanol + 0.02% glucose or the 0.02% glucose medium. Thus, as assayed by growth in synthetic ethanol medium, the *rpl1b* null mutants are grande, while the ethidium bromide controls are petite.

In the experiment above, a trace amount of glucose was present. To demonstrate that the *rpl1b* deletion strains could grow on ethanol in the complete absence of glucose, the cells (12A, B, C, and D) were grown to stationary phase in YNG 2% ethanol 0.02% glucose, then diluted from this medium into YNB 2% ethanol (i.e., without any glucose), with an initial concentration of about 5 × 10^5^ cells/mL. In this experiment, 12A, B, and D again grew to about 1 × 10^8^ cells/mL. Spore clone 12C failed to grow significantly in this experiment, but in my experience the transition to YNB medium with the ethanol carbon source is challenging, and it is not uncommon for a culture to fail to grow. 12C is wild-type for *RPL1B*, so both *rpl1b* mutants (12B and 12D) did grow to a stationary phase with ethanol as the sole carbon source. This experiment could not be controlled perfectly, because the petite controls could not be grown in ethanol medium. Instead, the petite control strains were grown to a stationary phase in YNB 2% glucose medium, and diluted into the YNB ethanol medium. These showed no detectable growth.

To assay the *rpl1b* phenotype in additional strains and backgrounds, I crossed spore clone BF534-12D (*KANR::rpl1b*, and monosomic for chromosome VII) to the non-isogenic strain GZ239 [10] to create diploid BF535, sporulated, and dissected six complete tetrads. Sporulation was efficient, and spore viability was high. In this cross, all six tetrads segregated 2:2 for G418 resistance, and for slow growth, which perfectly co-segregated. All 24 spore clones grew on YEP+ethanol (i.e., none were petite). The “bone white” appearance noted above for BF534-12B and 12D also appeared in many spore clones, but did not always segregate 2:2 and did not co-segregate with G418 resistance. Tetrad 1 was analyzed by PCR, which showed that spore clones B and D, which were the two G418-resistant, slow-growing clones, contained *KANR::rpl1b*, but did not contain WT *RPL1B* (Appendix A). In contrast, spore clones A and C, which were G418-sensitive, contained WT *RPL1B* but not *KANR::rpl1b* (Appendix A). These four spore clones were analyzed as above for the ability to grow in YNB + ethanol (no glucose). All four spore clones were able to grow to high density with ethanol as the sole carbon source (Figure 3), while their EtBr-generated petite controls failed to grow.

### 2.4. rpl1b Mutants Do Not Have a Relative Growth Defect in Ethanol

Although the experiments above show that *rpl1b* mutants can grow on ethanol as a carbon source, it is still possible that they grow relatively slowly on ethanol (i.e., relative to glucose, and relative to a wild type). To test this, I measured doubling times for eight strains from two tetrads, strains BF535-1a, 1b, 1c, 1d, 4a, 4b, 4c, and 4d. Four of the eight were *RPL1B*, and four were *rpl1b*. Doubling times were measured in YEP+glucose, and in YEP+ethanol (Materials and Methods). Results are shown in Table 1. The average ratio of the doubling times (ethanol T_D_/glucose T_D_) was 1.51 for the WT strains, and 1.56 for the *rpl1b* strains. These ratios are not significantly different (*p* = 0.81). Thus, the *rpl1b* strains do not seem to be at any relative disadvantage in ethanol medium.

Segev and Gerst [6] found that the ribosomal protein mutants *rpl2b*, *rps11a*, and *rps26b* were also petite. I recovered these mutants from the deletion set (in the case of *rps11a*, in both mating types), and the *KANR*::ribosomal protein replacement was assayed by PCR. All four strains showed the presence of the *KANR*::ribosomal protein replacement (Appendix A). In these cases, I did not do PCR reactions specific for the wild-type gene. These strains were grown in YNB 2% ethanol 0.02% glucose as above, with controls as above. All four deletion strains grew to about 1 × 10^8^ cells/mL in YNB 2% ethanol + 0.02% glucose, but not in 0.02% glucose (Figure 3). This shows that these strains are competent for respiration, and as such, are grande, not petite.

When the *KANR::rpl1b* strains were streaked for single colonies on either YEPD plates or YGlyE plates, fast-growing colonies could often be seen. One way to suppress the loss of *RPL1B* would be to provide a second copy of the paralog, *RPL1A*, which is on chromosome 16, and therefore, a reasonable hypothesis would be that the fast-growing variants might be chromosome 16 disomes. Chromosome 16 also carries *ARO7* (though it is genetically unlinked to *RPL1A*). To test this hypothesis, I picked two independent fast-growing variants of *KANR::rpl1b* strains, and crossed them to a *KANR::aro7* mutant from the deletion set. Seven tetrads were dissected for each cross, with perfect viability. Tetrads segregated 2:2 and 1:3 and 0:4 for slow growth–fast growth, consistent with the presence of a single-gene suppressor. Tetrads also segregated 2:2 and 1:3 and 0:4 for Aro-:Aro+, consistent with a second copy of *ARO7* in the *ARO7* parent. In both cases, 1:3 was the most common segregation. This is as expected if the *ARO7 RPL1A KANR::rpl1b* parents were disomic for chromosome 16. It is likely that, if an *rpl1b* strain were passaged repeatedly without single-colony selection for the slow-growers, it would accumulate a disomic copy of chromosome 16 as a suppressor.

## 3. Discussion

Here, I find that *rpl1b*, *rpl2b*, *rps11a*, and *rps26b* mutants all grow on ethanol; they are therefore grande and competent for respiration. Although the *rpl1b* mutant grows more slowly than the wild-type, this is true on glucose as well as on ethanol, and the mutant has no specific relative disadvantage on ethanol. There is no phenotypic evidence for the specialization of Rpl1b for respiration.

One crucial experimental difference between these experiments and those of Segev and Gerst, is that here, gold-standard control petite strains were made using ethidium bromide, and the *rpl1b* and *RPL1B* strains were compared to these petite controls. In contrast, Segev and Gerst [6] did not use any known control petite strains, and did not compare the growth of their *rpl1b* strains to that of a control petite.

For evaluating phenotype, Segev and Gerst [6] mainly relied on the observation that *rpl1b* strains grow somewhat more slowly on YPGly plates than on YPD plates. This is true here: *rpl1b* strains have a doubling time about 50% longer in ethanol than in glucose (Table 1); however, similarly, wild-type strains also grew more slowly on ethanol than on glucose. In my hands, there is little if any relative difference in the ability of *rpl1b* vs. WT strains to grow on ethanol versus glucose (Table 1). Here, but not in Segev and Gerst, strains were challenged to grow in synthetic media that contained either very little (0.02%) or no fermentable carbon source, and it was found that *rpl1b* strains were able to grow to high cell densities.

Segev and Gerst [6] also undertook gene expression/translation studies. However, it may be difficult to interpret the results of these studies in the absence of strains to control for disomy, since *rpl1b* strains accumulate a second copy of chromosome 16, and disomy is known to induce stress-related changes in gene expression. There are several other issues in interpreting Segev and Gerst’s studies on translation in *rpl1b* mutants. One issue is that the effects are small. A second issue is that it is not clear whether the finding of depletion of mitochondria-related proteins in *rpl1b* strains is correct. Segev and Gerst say “~40% of the proteins depleted from the *rpl1b* translatome in cells grown on glucose are functionally related to mitochondria (*n* = 21/54 depleted proteins.)”; however, Segev and Gerst give no definition, or list, or citation, for what proteins they consider “functionally related to mitochondria”. The critical 21 depleted proteins, “functionally related to mitochondria”, are re-listed here in Table 2. For these 21, there are 11 where the SGD Description includes the words “mitochondria” or “mitochondrial” or “respiration” or related words. The other 10 have no indication in their SGD descriptions of any mitochondrial function, and include, for instance, the cytoplasmic, glycolytic enzymes Eno1, Eno2, Gpm1, Pfk2, and Tdh2, and it is hard to see on what basis these glycolytic enzymes are included by Segev and Gerst in their statistical calculation as “functionally related to mitochondria”. For the 11 genes where the SGD Description does include “mitochondria”, etc., 5 seem to have a mitochondrial-specific function (*ATP1*, *GCV2*, *KGD1*, *SSC1*, *UTH1*), while the remaining 6 have both mitochondrial and non-mitochondrial functions (e.g., *ALA1*, *GRS1*, and *VAS1*, the cytoplasmic and mitochondrial tRNA-synthases for alanine, glycine, and valine). Similar comments apply to the proteins “functionally related to mitochondria” depleted from cells grown on glycerol. Thus, the claim of the specific depletion of mitochondrial-related proteins in the translatome of *rpl1a* may be exaggerated. In general, one cannot do a rigorous statistical test of depletion of proteins “functionally related to mitochondria” unless there is an objective list of such proteins, where the list to be used existed and was identified prior to and independently of the collection of data [11].

*rpl1b* strains rapidly accumulate growth suppressors, which may be predominantly a second copy of chromosome 16, which carries the paralog, *RPL1A*. Careful attention must be paid to such suppressors in phenotypic studies. The choice of a suppressed or non-suppressed isolate could greatly affect the growth phenotype, without informing on the growth effect of the underlying mutation, *rpl1b*.

The limitations of this study are as follows. The *rpl1b* mutant strains studied were all derived from the *MATalpha rpl1b* strain of the yeast deletion set. This strain is irregular in being a disome for chromosome VII, with both chromosome copies carrying the *KANR::rpl1b* mutation. This irregularity cannot help but cast some doubt on the genotype of the strain. The strains originally used by Segev and Gerst were requested from Dr. Gerst, but he did not respond. However, many PCR experiments, involving a variety of primer pairs, were performed on the parental strain, and on its descendant BF534-12d, and on four of its descendants from the BF535 diploid. All these PCR reactions, with no exceptions, show the presence of the *KANR::rpl1b* disruption, and the absence of the wild-type *RPL1B* gene, in the slow-growing, G418-resistant strains. Furthermore, slow-growth co-segregated 2:2 with G418 resistance in the six tetrads from cross BF535, with the *KANR::rpl1b* disruption confirmed by PCR in one tetrad, and none of these 24 strains were petite. Furthermore, these slow-growing strains are suppressed by a second copy of chromosome 16, which carries *RPL1A*. Furthermore, all three of the other ribosomal protein mutant strains in question did carry the expected deletions as assayed by PCR, and did also grow on ethanol.

## 4. Materials and Methods

### 4.1. Yeast Strains

Strains are described below, and in the text.

Mutant strains were recovered from the yeast deletion set [12]. Deletions were confirmed by assaying resistance to G418, and by PCR. As noted, mutant strains were crossed to the isogenic strains BY4741 (*MATa his3 leu2 met15 ura3*) or BY4742 (*MATalpha his3 leu2 lys2 ura3*) or to the related but not isogenic strain GZ239 [10]. GZ239 has five polymorphisms compared to BY4741/BY4742. These are wild-type *HAP1*, and the mitochondria-stabilizing alleles *SAL1*, *CAT5-91M*, *MIP1-661T MKT1-30G* alleles of the strain UCC8376 [9,10]. These polymorphisms explain much of the variation in mitochondrial maintenance in laboratory strains [9]. Consequently, in a cross between the BY4741 background and GZ239, a large number of mitochondrial-relevant genotypic combinations can be surveyed.

### 4.2. Media and Growth Conditions

YEPD (aka “YPD”) was 2% peptone, 1% yeast extract, 2% glucose, with 2% agar when made as solid medium.

YGlyE was 2% peptone, 1% yeast extract, 0.1% glucose, 1% glycerol, 2% ethanol, with 2% agar when made as solid medium.

YEP+ethanol was 2% peptone, 1% yeast extract, 2% ethanol.

YNB (yeast nitrogen base) was 5g/L ammonium sulfate, 1.7 g/L yeast nitrogen base without ammonium sulfate and without amino acids. Amino acids or bases were added to 50 mg/L (histidine, uracil, lysine, methionine) or to 150 mg/L (leucine). Carbon sources (glucose or ethanol) were added as indicated.

Cells were grown at 30 °C.

### 4.3. PCR Primers

KanR R longAAACGTGAGTCTTTTCCTTACCCKanR R shortGCGACAGTCACATCATGCCCRPL1b upUTR FGCATTACGTTATTTGGTAACCTCRPL1b ORF RGTTTGGCAACTTCAAAGAACCRPL2B FTGTTGGATACTAAGCAGTTCCCRPL2B RGTAATATGTCAATTTCAATAAGTCCCRPS11a FCCCAAAATTCATACTGTCTCACTCRPS11a RCTTTATCAAGCCTTCGGCACCRPS26b FTGATATTCATACCGACACTATAATCCACRPS26b RAGCCATTCTCTTGATAGCCTTATCC

The positions of the PCR primers in the genes of interest are shown in Appendix A.

### 4.4. Growth Rate Measurements

For each strain, a single colony was picked from a streak on a YEPD plate. For *rpl1b* mutants, the colony chosen was a small colony, but representative of the vast majority of the colonies on the plate (i.e., the few large colonies were avoided). Strains were grown in YEP–ethanol liquid to saturation, diluted 100-fold in the same medium, then grown to saturation again to adapt all strains to ethanol. For each strain, a sample was sonicated, and the number of cells/mL was accurately counted with a Z2 Beckman-Coulter Channelyzer. Typically, cells/mL at saturation was around 2 × 10^8^ cells per ml. Cells were diluted into fresh medium (for each strain, into YEPD, and also into YEP–ethanol) at a calculated concentration of 2.0 × 10^6^/mL. Cells were grown in 5 mL of liquid medium in 20 mL tubes, slanted, in a roller, in a 30 °C incubator. The number of cells per ml was followed with time using the Z2 Beckman-Coulter Channelyzer. Cells were diluted as necessary to maintain cells in continuous log phase growth, at cell concentrations between 5 × 10^5^/mL and 5 × 10^7^/mL. For most strains, the growth experiment lasted about 30 h, but the experiment lasted longer for the two slowest growing strains (535-4c and 535-4d), and lasted nearly 70 h for 535-4c in ethanol.

Medians of the cell size distributions were also recorded; however, within a few hours after the initial inoculation, these showed no significant change with time, suggesting that the cells were in an equilibrium state. Different strains had only moderately different cell sizes from each other, and sizes in ethanol were smaller than in glucose, as expected. (Since the cells pre-grown in ethanol were small, there was an increase in cell size after these were inoculated into glucose medium, as expected, but the new equilibrium was reached shortly.)

### 4.5. Generation of “Gold Standard” Petite Variants

To generate petite variants, a shallow well (about 100 microliters in volume) was dug in the center of a YEPD plate using a sterile platinum loop (but the well did not extend all the way through the agar). Then, 100 microliters of 10 mg/mL ethidium bromide was pipetted into the well. After all liquid had been absorbed, yeast was streaked on the quadrants of the plate, with the streak extending from the outer edge of the plate to the edge of the ethidium bromide well. After about two days of incubation at 30 °C, small, white, single colonies were visible close to the ethidium bromide well (and at this time, a red halo of ethidium bromide could be seen surrounding the well). Several of these small, white colonies were picked, and re-streaked for single colonies on YEPD plates. Single colonies from the YEPD streak were chosen, and tested for their ability to grow on YGlyE petite indicator plates, in comparison to their growth on YEPD plates. Colonies that grew well on YEPD, but very badly on YGlyE, were considered petite variants and were stored. Typically, these had a bone-white appearance. None of these was able to grow detectably in synthetic medium with ethanol as the sole carbon source.

## Figures and Tables

**Figure 1 ijms-25-11553-f001:**
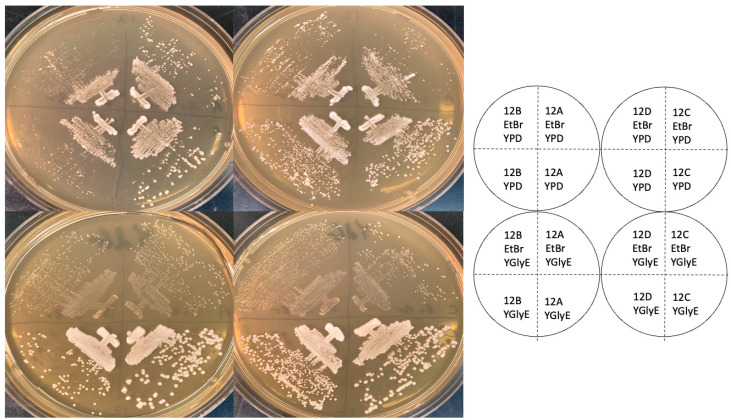
Growth phenotypes on YEPD and YGlyE indicator plates. Spore clones 12A, B, C, and D were streaked on YEPD plates (top two plates, “YPD”) and grown for two days, or YGlyE petite indicator plates (bottom two plates) and grown for four days. On the top half of each plate are control petite strains generated using ethidium bromide, to show a genuine petite phenotype, and on the bottom half, are the experimental strains.

**Figure 2 ijms-25-11553-f002:**
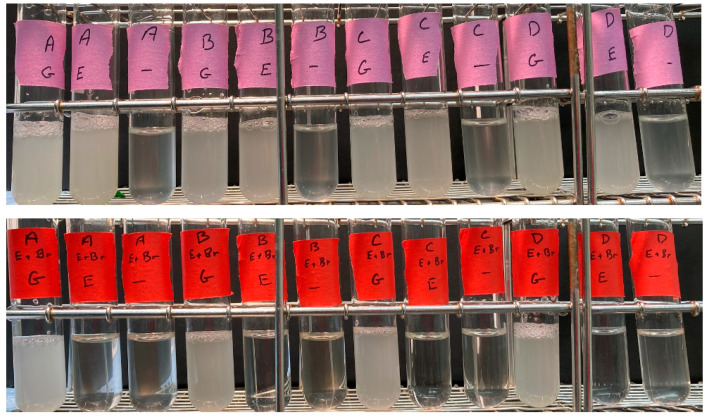
Growth phenotypes of spore clones from BF534 tetrad 12 in synthetic medium. Top row: Spore clones BF534-12A, B, C, and D were grown in yeast nitrogen base synthetic medium with a carbon source of 2% glucose (“G”) or 2% ethanol plus 0.02% glucose (“E”) or 0.02% glucose (“-”). Bottom row: As above, but the ethidium bromide-derived control petite versions were used.

**Figure 3 ijms-25-11553-f003:**
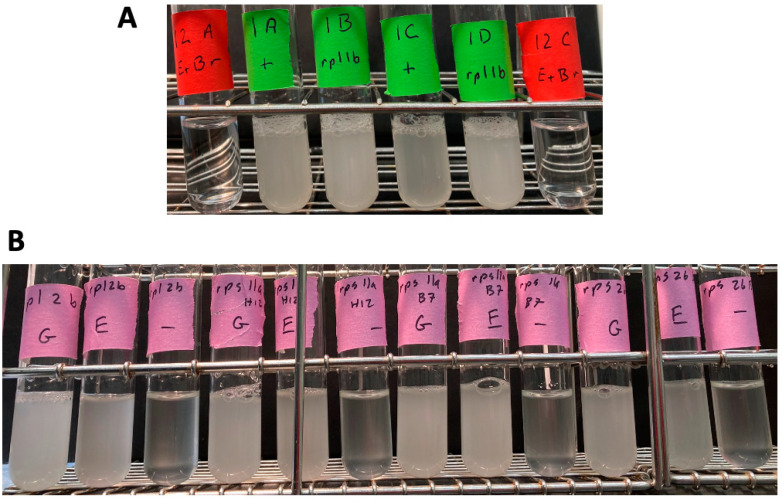
Growth of mutant spore clones in synthetic medium plus ethanol. (**A**) Spore clones 1A (WT), 1B (*rpl1b*), 1C (WT), and 1D (*rpl1b*) from BF535 tetrad 1 were grown in yeast nitrogen base with 2% ethanol and no other carbon source. Ethidium bromide generated petites (BF534-12A and BF534-12C, both containing *RPL1B*) are shown as controls. (**B**) The *rpl2b*, *rps11a* (in both mating types) and *rps26b* mutants from the deletion set were grown in yeast nitrogen base synthetic medium with 2% glucose (“G”), or 2% ethanol plus 0.02% glucose (“E”), or 0.02% glucose (“-”). All four strains grew on ethanol.

**Table 1 ijms-25-11553-t001:** Growth rates of WT and *rpl1b* in ethanol and glucose media.

Strain	*RPL1B*	T_D_ Ethanol (h)	T_D_ Glucose (h)	Ratio (E/G)
535-1a	+	2.43	1.61	1.51
535-1b	−	2.51	1.83	1.37
535-1c	+	2.51	1.64	1.53
535-1d	−	3.24	2.16	1.50
535-4a	+	2.81	1.64	1.71
535-4b	+	2.09	1.59	1.31
535-4c	−	5.38	2.60	2.07
535-4d	−	3.55	2.71	1.31

Strains were grown in YEP+ethanol or YEP+glucose (Materials and Methods), and doubling times were measured. The doubling time in each medium (T_D_) is given in hours. The ratio of the doubling time in ethanol divided by the doubling time in glucose is given.

**Table 2 ijms-25-11553-t002:** Functions of proteins “functionally related to mitochondria”.

Protein	“Mito” Function in SGD Description	Function Specific to Mitochondria	Function (Abbrev. from SGD)
Ala1	Yes	No	Cytoplasmic and mitochondrial alanyl-tRNA synthase
Ald6	Yes	No	Cytosolic aldehyde dehydrogenase; localizes to the mitochondrial outer surface upon oxidative stress
Atp1	Yes	Yes	Alpha subunit of the F1 sector of mitochondrial F1F0 ATP synthase
Cdc48	No	No	AAA ATPase with protein-unfoldase activity
Eno1	No	No	Converts 2-phosphoglycerate to phosphoenolpyruvate; glycolysis
Eno2	No	No	Converts 2-phosphoglycerate to phosphoenolpyruvate; glycolysis
Erg6	Yes	No	Delta(24)-sterol C-methyltransferase; converts zymosterol to fecosterol in ergosterol biosynthesis
Fas1	No	No	Beta subunit of fatty acid synthetase
Gcv2	Yes	Yes	P subunit of the mitochondrial glycine decarboxylase complex
Glt1	No	No	NAD(+)-dependent glutamate synthase
Gpm1	No	No	Phosphoglycerate mutase; glycolysis and gluconeogenesis
Grs1	Yes	No	Cytoplasmic and mitochondrial glycyl-tRNA synthase
Kgd1	Yes	Yes	Subunit of mitochondrial alpha-ketoglutarate dehydrogenase
Pfk2	No	No	Beta subunit of phosphofructokinase, glycolysis
Ssa2	No	No	HSP70 ATP-binding protein involved in protein folding
Ssc1	Yes	Yes	Hsp70 ATPase; component of import motor of the translocase of the inner mitochondrial membrane
Tcb3	Yes	No	Lipid-binding ER tricalbin involved in ER-plasma membrane tethering
Tdh1	No	No	Glyceraldehyde-3-phosphate dehydrogenase, glycolysis
Ura2	No	No	Catalyzes first two steps in pyrimidine synthesis
Uth1	Yes	Yes	Mitochondrial inner membrane protein
Vas1	Yes	No	Mitochondrial and cytoplasmic valyl-tRNA synthetase

Protein: Common name of protein. “Mito” function in SGD description: Whether or not the Saccharomyces Genome Database description of the gene’s function includes the words “mitochondria”, “mitochondrial”, or “respiration”, or other closely related term (e.g., “oxidative phosphorylation” would be considered a closely-related term). Function specific to mitochondria: a somewhat subjective evaluation of whether the function specified in the SGD description is a function specific to mitochondria.

## Data Availability

All relevant data are contained within the manuscript and the Appendix A. Strains are available upon request from BF.

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
