# Peer review of "The Yeast Ribosomal Protein Rpl1b Is Not Required for Respiration"

_ijms, 2024, doi:10.3390/ijms252111553_

Round 1

Reviewer 1 Report

Comments and Suggestions for Authors

This manuscript refutes the main conclusions described in a 2018 publication from Segev and Gerst in J Cell Biol 217:117, as the author now finds that a foundational premise of the earlier work is incorrect. Segev and Gerst based their studies on their observation that deletion of only one of the two paralogs encoding any of four ribosomal proteins in the yeast Saccharomyces cerevisiae caused failure of mitochondrial function leading to a “petite” phenotype. These observations were the basis for suggesting that some paralogs formed ribosomes with distinct functions, a surprising result given that most of the paralogs encode proteins with identical sequences. Dr. Futcher now reports that deletions of the paralogs noted by Segev and Gerst do not lead to mitochondrial failure and are not associated with the petite phenotype, seriously undermining the conclusions reported by Segev and Gerst. The current study reports careful and extensive analysis of rpl1b with more cursory descriptions of the other three genes described by Segev and Gerst, but the conclusions are the same in all four cases. Notably, the data presented by Segev and Gerst did not include a control petite strain, and the growth they show in their publication is much more robust on non-fermentable carbon sources than would be expected for actual petite strains. The current manuscript includes these controls and examines multiple non-fermentable carbon source growth conditions, providing high confidence in the conclusion that the deletion strains tested are not petite. Importantly, the studies include segregation analysis in two genetic backgrounds demonstrating the validity of the inferences and suggesting how the previous error could have occurred. The data are of high quality, confidence in the conclusions is high, and the impact of the study is to correct a significant and often cited error in the published literature.

No major concerns with the manuscript were identified but several minor issues should be addressed.

1) Most significantly, the data showing segregation of phenotypes with the deletion of rpl1b are a crucial element of this manuscript (and a major deficiency in the Segev and Gerst publication). Dr. Futcher follows a careful strategy of using PCR primers that anneal to various positions within the WT RPL1b gene and the KANR marker used to replace it to correlate genotype with phenotype, but although the legend to Supplementary Figure 1A indicates that the sequences of the primers are to be found in the materials and methods section, I was unable to find them. Ideally, the figure would include a cartoon mapping the positions of the primers so that the reader can easily follow which primers are being used to test which structures of the genes.

2) While the segregation ratios described clearly indicate that suppression is likely to have occurred through formation of a 1n+1 disome in the deletion collection, it might be helpful to non-experts to make an additional cartoon indicating how this explains the PCR and phenotype results.

3) The references are both numbered and in alphabetical order in the bibliography but are called by author name in the text, making the numbering unnecessary and at least momentarily confusing.

Author Response

Reviewer Comment 1) Most significantly, the data showing segregation of phenotypes with the deletion of rpl1b are a crucial element of this manuscript (and a major deficiency in the Segev and Gerst publication). Dr. Futcher follows a careful strategy of using PCR primers that anneal to various positions within the WT RPL1b gene and the KANR marker used to replace it to correlate genotype with phenotype, but although the legend to Supplementary Figure 1A indicates that the sequences of the primers are to be found in the materials and methods section, I was unable to find them. Ideally, the figure would include a cartoon mapping the positions of the primers so that the reader can easily follow which primers are being used to test which structures of the genes.

Response:  First, I thank the Reviewer for his/her careful and thoughtful review.  He/she is correct that I failed to include the primer sequences.  In the modified manuscript the primer sequences are included, and there is a new supplementary figure that shows the position of the primers on the RPL1B gene and on the rpl1B::G418 gene.  I believe this is what the reviewer wanted, and these are good additions.

Reviewer Comment 2) While the segregation ratios described clearly indicate that suppression is likely to have occurred through formation of a 1n+1 disome in the deletion collection, it might be helpful to non-experts to make an additional cartoon indicating how this explains the PCR and phenotype results.

Response:  It is hard for me to judge how many readers will have trouble understanding the segregation of a disomic chromosome in a cross.  But I take the reviewer's word for it that an explanation would be helpful.  There is now a new supplementary figure showing how the disomic chromosome will segregate in a cross, and relating that to tetrad 534-12.  I have not given all the details as to how the PCR and growth phenotypes will be affected, but it would take a lot of space to do that, and once the segregation and genotypes are made clear, as in this new supplementary Figure, it should be easy for almost anyone to work out the phenotypic consequences.

Reviewer Comment 3) The references are both numbered and in alphabetical order in the bibliography but are called by author name in the text, making the numbering unnecessary and at least momentarily confusing.

Response.  I contacted the Journal about the format of the references.  I was told to leave the formatting as is, and the Journal will make adjustments as necessary.

In summary, all of the reviewer's suggestions were good, and I have improved the manuscript by making the changes requested for the first two comments, and the Journal will take care of the last comment, about formatting the references.

Reviewer 2 Report

Comments and Suggestions for Authors

The concept of specialized ribosomes, that not all ribosomes are functionally equivalent, has garnered a lot of enthusiasm but also some skepticism. Because of the high impact of this concept on our understanding of gene expression, it is important that the basis for establishing ribosome specialization is rigorously assessed. In 2018, Sergev and Gerst published a much-cited paper supporting the idea of ribosome specialization. Their conclusion was based on the analysis of yeast paralogs of ribosomal protein genes. For example, they concluded that RPL1B but not RPL1A was required for respiration, despite the fact that the two paralogs are identical in amino acid sequence. In this brief report, the author provides evidence that mutants deleted individually of four different ribosomal protein genes (RPL1B, RPL2B, RPS11A, and RPS26B) are competent for respiration. That is, an rpl1b mutant, for example, is not petite and does not specifically affect mitochondrial function. These are the same genes studied in the Sergev and Gerst paper, but the conclusions are contrary. Thus, this manuscript raises doubts about the conclusions in the Sergev and Gerst paper and is important for the field to consider when building a case for ribosome specialization. In general, the genetic experiments appear well-executed. However, there are several minor concerns. Because the results of this manuscript are contrary to published work, the onus is on the author to be rock solid on these new results and conclusions.

Points to address:

1. The bulk of the work is based on an rpl1b mutant. The author chose to use an existing mutant from the yeast knockout collection and discovered the problem of aneuploidy in this collection, especially common among the ribosomal protein deletion strains. The author uses a spore clone obtained from crossing the strain with WT to eliminate the extra ChrVII for subsequent work. The concern here is that the spore clones obtained from this cross may contain additional suppressing mutations. This concern derives from the fact that the original strain was apparently disomic for ChrVII, carrying the rpl1b deletion. As he notes later, rpl1b mutants commonly display aneuploidy for ChrXVI, which carries the RPL1A paralog. What would be the driver for disomy for the rpl1b deletion in the starting strain? Could there be another mechanism of suppression that the author has not identified that could confound his interpretation? The manuscript would be much more straightforward to follow and the interpretation more solid if the author had derived the rpl1b depletion de novo.

2. Lines 116-119, an experiment was done to demonstrate that rpl1b deletion strains could grow on ethanol in the complete absence of glucose. Oddly, spore clone 12C failed to grow. The author ascribes this to the vagaries of culture growth. However, this is a trivial experiment and to rule out any other possibilities shoulb be repeated.

Author Response

Reviewer 2 Comment 1. The bulk of the work is based on an rpl1b mutant. The author chose to use an existing mutant from the yeast knockout collection and discovered the problem of aneuploidy in this collection, especially common among the ribosomal protein deletion strains. The author uses a spore clone obtained from crossing the strain with WT to eliminate the extra ChrVII for subsequent work. The concern here is that the spore clones obtained from this cross may contain additional suppressing mutations. This concern derives from the fact that the original strain was apparently disomic for ChrVII, carrying the rpl1b deletion. As he notes later, rpl1b mutants commonly display aneuploidy for ChrXVI, which carries the RPL1A paralog. What would be the driver for disomy for the rpl1b deletion in the starting strain? Could there be another mechanism of suppression that the author has not identified that could confound his interpretation? The manuscript would be much more straightforward to follow and the interpretation more solid if the author had derived the rpl1b depletion de novo.

Response:  First, I thank the reviewer for his careful and thoughtful review, which does get to two (relatively) weaker points in the research. 

If I knew then what I know now, I would have made the rpl1b deletions de novo, and of course it would have made the manuscript much more straightforward.  But I didn't know, and this is not a research area for me--I have no grant, and no possibility of a grant, to do this.  I was working on a side project with what was available, and (as I said in the manuscript) I was already using the rpl1b segregants in a different kind of experiment, which relied on their supposed petite phenotype.  So I was some distance down a path before I realized there was no petite phenotype.  I will say two kinds of things here.  First, it is really pretty certain that there is no other mechanism of suppression, because spore clone 534-12D was crossed to another strain, GZ239, and six complete tetrads were dissected.  All of these tetrads segregated 2:2 for slow growth, and G418 resistance perfectly co-segregated with this slow growth, and none of the 24 strains was petite.  This is incompatible with the idea of a suppressor (unless of course the suppressor is incredibly tightly linked).  This kind of tetrad analysis is the classic way (I would say the Gold Standard way) of asking if there is a suppressor, and here, by this analysis, the answer is no.  PCR analysis was done on one of these tetrads, and showed that the rpl1B::G418 was co-segregating with slow growth and G418 resistance, while RPL1B was co-segregating with fast growth.  Also of course, the results were the same with rpl2b, rps11a, and rps26b.  This is all in the paper, and readers can make up their own minds about rigor.  The second thing to be said is that ultimately, in the extreme, one wants the actual rpl1b deletion strain made by Segev and Gerst.  I had some correspondence with Dr. Gerst, and requested the strains, but he did not send them (he didn't say he would or he wouldn't, but in fact he didn't send them).  So getting these strains was just out of my hands. 

Reviewer Comment 2. Lines 116-119, an experiment was done to demonstrate that rpl1b deletion strains could grow on ethanol in the complete absence of glucose. Oddly, spore clone 12C failed to grow. The author ascribes this to the vagaries of culture growth. However, this is a trivial experiment and to rule out any other possibilities shoulb be repeated.

Response:  As the reviewer says, it is a trivial experiment, and so of course I did try again with spore clone 12C, by re-inoculating from the same starter culture, and it did grow when I tried again.  So the experiment was indeed repeated for this spore clone, and gave the expected result (i.e., growth for 12C).  But this wasn't part of the full experiment with all the strains and all the controls, so what is shown is what happened the first time (i.e., non-growth for 12C) in the full experimental context.

Reviewer 3 Report

Comments and Suggestions for Authors

This work done by Bruce Futcher expressed the objection of the results rpoeted by Segev and Gerst in 2018 regarding the effect of loss of functional ribosomal proteins Rpl1b on respiration of S. cerevisiae. This is something worth to present in public. 

Line54: ‘This suggests that the MATalpha strain in the deletion set is a disome for chromosome VII’

--Are you suggesting that all MATalpha strains have a disome of chromosome VII or is it only proven in the strain you analyzed? If the yeast deletion collection all have a disome of chromosome VII, it must affect the results of many works. Is it known among the community of yeast researchers or in literatures?

Line 105: the rpl1b null mutants

 --Italicize rpl1b

Line 169: This shows that these strains are competent for respiration, and as such, are grande, not petite.

--Does the author have this result to show?

Line251:

Which ‘four’ means in the description of ‘all four of the other ribosomal protein mutant strains’ ? Clarify. 

Figure 1 and its description: 

‘BF534-12B and 12D were both slow-growing on YEPD, as expected for an rpl1b deletion, or for a petite. They both also formed the bone-white colonies typical of a petite, instead of the creamier colonies of a grande’

--Looking at the Figure 1, I agree 12B forms smaller colonies, but 12D does not show clear difference with 12C. 

-- The author labeled YPD in this figure but the legend stated YEPD. Need consistency.

Figure 3B:

 The left three are rpl2b. The legend goes rpl1b.

Supplementary Fig. 1B:

‘Segregation of the wild-type RPL1B gene in BF534-12’

Are these same with BF535-1a, 1b, 1c, 1d in Table 1? If so, be consistent.

Author Response

Reviewer 3 Comment 1.  Line54: ‘This suggests that the MATalpha strain in the deletion set is a disome for chromosome VII’

--Are you suggesting that all MATalpha strains have a disome of chromosome VII or is it only proven in the strain you analyzed? If the yeast deletion collection all have a disome of chromosome VII, it must affect the results of many works. Is it known among the community of yeast researchers or in literatures?

Response.  I am only suggesting that this particular strain is a disome.  Of course, many strains from the yeast deletion set have been analyzed, and most of the strains, including most of the MATalpha strains, are not disomes.  On the other hand, it is not rare to find that a strain from the deletion set is a disome--I think I have heard estimates that around 5% (???? not sure of the number) may be disomes.

Reviewer 3 Comment 2.  Line 105: the rpl1b null mutants

 --Italicize rpl1b

Response:  OK.  I agree it should be in italics--but as far as I can see in my version, it is in italics.  I am not sure why it is apparently different from the Reviewer's version.  Perhaps I corrected it after submission.

Reviewer 3, Comment 3.  Line 169: This shows that these strains are competent for respiration, and as such, are grande, not petite.

--Does the author have this result to show?

Response:  Yes, I believe that Figure 1, and Figure 2, and Figure 3 show this.  Perhaps I am not understanding what the reviewer is asking.

Reviewer 3, Comment 4.  Line251:

Which ‘four’ means in the description of ‘all four of the other ribosomal protein mutant strains’ ? Clarify. 

Response:  Sorry, the reviewer is correct, there are four in total.  This line has now been corrected to say "Furthermore all three of the other ribosomal protein . . .", and the other three are Rpl2b, Rps11a, and Rps26b.

Reviewer 3 Comment 5.  

Figure 1 and its description: 

‘BF534-12B and 12D were both slow-growing on YEPD, as expected for an rpl1b deletion, or for a petite. They both also formed the bone-white colonies typical of a petite, instead of the creamier colonies of a grande’

--Looking at the Figure 1, I agree 12B forms smaller colonies, but 12D does not show clear difference with 12C. 

-- The author labeled YPD in this figure but the legend stated YEPD. Need consistency.

Response.  Well, I think the colonies are a little smaller for 12D than 12C.  And I have looked at them on a lot of streaks on a lot of plates, and they are consistently smaller.

I added to the Fig. 1 legend that the "YPD" label on the image means YEPD.

Reviewer 3 Comment 6.  Figure 3B:

 The left three are rpl2b. The legend goes rpl1b.

Supplementary Fig. 1B:

‘Segregation of the wild-type RPL1B gene in BF534-12’

Are these same with BF535-1a, 1b, 1c, 1d in Table 1? If so, be consistent.

Response:  With regard to Fig. 3, the Reviewer is correct, well-spotted, thank-you.  The Legend should say "rpl2b", not "rpl1b".  I have corrected it.

With respect to Supplementary Fig. 1B, the Reviewer is not correct.  BF534, and BF535, are different crosses, and the Legends are correct.  In fact, it is important to realize that these strains are NOT the same.  It is further evidence that the phenotypes are really the direct phenotypes of the rpl1b::G418 allele, and that no suppressor is involved.  See my discussion with Reviewer 2.